# Topology-aware Neural Flux Prediction Guided by Physics

**Haoyang Jiang** [1]   **Jindong Wang** [1]   **Xingquan Zhu** [2]   **Yi He** [1]

## Abstract

Graph Neural Networks (GNNs) often struggle in preserving high-frequency components of nodal signals when dealing with directed graphs. Such components are crucial for modeling flow dynamics, without which a traditional GNN tends to treat a graph with forward and reverse topologies equal. To make GNNs sensitive to those high-frequency components thereby being capable to capture detailed topological differences, this paper proposes a novel framework that combines 1) explicit difference matrices that model directional gradients and 2) implicit physical constraints that enforce messages passing within GNNs to be consistent with natural laws. Evaluations on two real-world directed graph data, namely, water flux network and urban traffic flow network, demonstrate the effectiveness of our proposal. The code for this paper is available at https://github.com/HaoyangJiang-WM/PhysicsNFP.

## 1. Introduction

Directed graphs are frequently used to model various physical and engineering systems, due to their strength in capturing spatial dependencies and complex interactions between components. Graph Neural Networks (GNNs) have emerged as powerful tools for modeling such graphs, particularly in applications like water flux prediction and traffic flow analysis (Kratzert et al., 2021; Jin et al., 2023). However, recent studies (Kirschstein & Sun, 2024) have revealed a critical limitation, that GNNs often struggle in modeling physics-based flow dynamics due to their insensitivity to *edge directions*.

In real systems, flow dynamics follow strict physical laws, where local and rapid changes, *e.g.*, turbulent eddies, sharp

flow transitions, or abrupt flux variations, only propagate in specific directions (Sagaut, 2005; LeVeque, 2002; Canuto, 2007). Yet, GNNs typically yield similar performance whether the original edge directions are maintained, reversed, or randomly perturbed. This directional insensitivity mainly results from the message-passing mechanism in GNNs, which implicitly acts as a low-pass filter (Kesting & Treiber, 2013; Sagaut, 2005). While this filtering enables GNNs to capture low-frequency patterns, such as seasonal trends in river networks, it suppresses high-frequency variations that arise from rapid or local changes (Sun et al., 2022; Bo et al., 2021; Hoang et al., 2021).

In this paper, we mainly explore two key research questions: *i) why are GNNs insensitive to edge directions and ii) how can their directional awareness be improved*.

We hypothesize that the low-pass filtering nature of message passing is the main cause of this limitation. To validate this, we formulate an inverse problem for flux prediction in river networks, where the task is to infer upstream fluxes based on downstream observations. This ill-posed setup leads to instability by amplifying high-frequency components (Fisher et al., 2020; Ferrari et al., 2018), where small numerical errors can result in significant variations in inferred upstream conditions, making the problem highly sensitive to local flux changes. Yet, standard GNNs fail to capture these amplified high-frequency signals, resulting in poor performance when modeling directional dependencies.

To overcome these challenges, we propose a novel *physics-guided neural flux prediction* (PhyNFP) framework that integrates physical laws into GNN training, preserving high-frequency components for better flow dynamics modeling. Our framework has two main components: 1) At local level, PhyNFP replaces traditional adjacency matrices with *discretized difference matrices*, which encode local variations and directional dependencies between nodes. These matrices capture directional gradients, allowing the GNN to retain high-frequency information and distinguish flow directions. 2) At global level, PhyNFP incorporates physical equations that describe flow dynamics, *e.g.*, conservation of momentum, directly in GNN training. This physics-guided regularization ensures that predictions remain consistent with underlying physical principles. Note, our PhyNFP framework is generalizable in the sense that different physical

---

[1]Department of Data Science, William & Mary, Williamsburg, VA, USA [2]College of Engineering & Computer Science, Florida Atlantic University, Boca Raton, FL, USA. Correspondence to: Dr. Yi He <yihe@wm.edu>.

*Proceedings of the 42nd International Conference on Machine Learning*, Vancouver, Canada. PMLR 267, 2025. Copyright 2025 by the author(s).

equations can be adopted for various types of flow networks. We evaluate `PhyNFP` on two real-world datasets, implementing the Saint-Venant equations for river networks and the Aw-Rascle equations for traffic networks. Experimental results demonstrate that `PhyNFP` enhances GNN performance by improving sensitivity to directional dependencies and high-frequency dynamics. Furthermore, we validate our hypothesis regarding the inverse problem nature of the reversed topology by examining the model's behavior under perturbation in this setting.

**Specific contributions** in this paper include:

1. This is the first study to guide training of GNNs with physics information for flux prediction, in order to enhance their sensitivity to high-frequency components and edge directions.
2. An inverse problem is formulated to validate the low-pass filtering nature of GNNs, substantiating their incapability in capturing high-frequency components in nodal features hence insensitive to edge directions.
3. Empirical evaluations on two different directed networks demonstrate the effectiveness of our framework, which i) Outperforms its GNN competitors by 31.6% in the river dataset and 4.9% in the traffic dataset on average in flux prediction. ii) Uplifts the GNN sensitivity to edge directions by 96.5% in the river dataset and 79.9% in the traffic dataset.

## 2. Preliminaries

**Problem Statement.** Consider a directed graph $G(A) = (V, E)$ representing a flow network, where $A$ is the graph adjacency matrix and $A \neq A^\top$ in general. $V = \{v_i\}$ is the set of nodes, with each node $v_i$ associated with a vector $\mathbf{x}_i \in \mathbb{R}^{t \times p}$ that encodes the quantities of $p$ variables (*e.g.,* flux volume, density, and velocity) over $t$ time steps. We have $\mathbf{X} = [\mathbf{x}_1, \ldots, \mathbf{x}_{|V|}]^\top$ to denote the nodal feature matrix of $G$. Let $E \subseteq V \times V$ be the edge set, and $e_{ij} = (v_i, v_j) \in E$ represents an edge pointing from $v_i$ to $v_j$, associated with a vector $\mathbf{e}_{ij} \in \mathbb{R}^q$ that encodes physical quantities such as level difference or distance between nodes.

In this paper, we follow the prior study (Kirschstein & Sun, 2024) to frame the flux prediction task in supervised node regression. Specifically, our goal is to predict the *lead time* hours in the future, *i.e.,* predicting the flux volume at $t + n$ step for all nodes, where $n$ is a configurable prediction horizon. The ground-truth of all nodes is denoted by $\mathbf{y} \in \mathbb{R}^{|V|}$. Our objective takes the form:

$$\min_{\theta} \frac{1}{|V|} \sum_{v_i \in V} \ell\left(y_i, f(\mathbf{x}_i, \mathbf{e}_{ij}, A; \theta)\right),$$

where $\ell$ denotes the loss function (*e.g.,* MSE or RMSE), $f$ denotes the GNN model parameterized by $\theta$, $y_i \in \mathbf{y}$ is the true flux volume of node $i$ at step $t + n$.

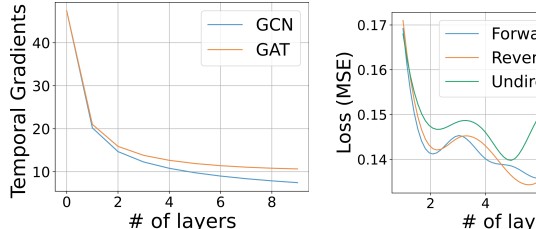

*Figure 1.* Left: Trends of temporal gradients *w.r.t.* the increasing number of message-passing layers. Right: MSE Trends of GCN in the original (Forward), inverse (Reverse), and undirected network settings *w.r.t.* the increasing number of message-passing layers.

### Technical Challenges

GNNs leverage neighborhood aggregation to yield node embeddings that harmonize information from both nodal features and graph topology. Denoted by $\mathbf{h}_i^{l+1}$ the embedding vector of node $v_i$ resulted from the $(l + 1)$-th message-passing layer, it is computed in a recursive form as follows.

$$\mathbf{h}_i^{l+1} = U_l\big(\mathbf{h}_i^l, \sum_{v_j \in \mathcal{N}_{\text{in}}[v_i]} M_l(\mathbf{h}_i^l, \mathbf{h}_j^l, \mathbf{e}_{ji})\big), \ \mathbf{h}_i^0 = \mathbf{x}_i \quad (1)$$

where $\mathcal{N}_{\text{in}}[v_i]$ denotes the *incoming* neighbors of $v_i$ in $G$, *i.e.,* $v_j$ is upstream of $v_i$. $U_l$ and $M_l$ denote the update and aggregation functions of the $l$-th layer, respectively.

This message-passing process leads to the smoothing effect because it inherently acts as a low-pass filter, which encourages similar embeddings of neighboring nodes and attenuates high-frequency components (Sun et al., 2022; Bo et al., 2021). Denoted by $\Delta \mathbf{x_i} = (1/t) \sum_{s=1}^{t} (\mathbf{x_i}[s] - \mathbb{E}(\mathbf{x_i}))^2$ the temporal gradient of each node $v_i$, which represents the rate of change of $p$ variables between consecutive time steps. Figure 1 (left) illustrates the temporal gradients of all nodes in the *river* dataset (details in Sec. 4), and how they change *w.r.t* GNN layers. We observe that the differences among the temporal gradients of these nodes and their embeddings diminish with more message-passing layers, validating that high-frequency components, as rapid variations of input node features, cannot be captured in their embeddings.

We further validate that GNNs are insensitive to edge directions due to their incapability to capture these high-frequency components. To wit, we set up an inverse problem of our prediction task. Specifically, in our original problem, information propagates downstream in both space and time, where each node embedding $\mathbf{h}_i$ depends on features from upstream nodes $v_j$, as indicated in Eq. (1). In its inverse problem, the edge directions are reversed, making $A^\top$ the graph adjacency. The task becomes ill-posed because it requires inferring upstream from downstream conditions, which incurs two issues. First, the upstream boundary conditions are lacking (Fisher et al., 2020), as downstream nodes do not contain sufficient information of upstream flow condi-

tions, making the mapping potentially one-to-many. Second, small numerical errors in the inference process can propagate and be amplified, leading to instability and sensitivity in the reconstructed upstream conditions. As a result, the solutions are non-unique with high-frequency noises amplified during inverse problem. This instability introduces high-frequency errors, causing small perturbations to result in drastically different inferred solutions.

Figure 2 (right) demonstrates the trends of prediction loss *w.r.t.* different numbers of forward (original), reverse, and undirected message-passing layers. Similar loss trends across all configurations indicate that while high-frequency attenuation increases GNN robustness by suppressing noise inherent in inverse problems, it simultaneously reduces the GNN sensitivity to changes in flow direction. This attenuation effect limits GNNs to capture complex flow dynamics, particularly in cases where distinguishing between forward and reverse flow directions is critical. More detailed analysis of the technical challenges are deferred to the supplementary material due to page limits.

## 3. Proposed Approach

To improve the directional awareness of GNNs, we propose `PhyNFP` that integrates explicit and implicit physical constraints. In this section, we first introduce discretized difference matrices, as explicit constraints, that model local gradient changes in Sec. 3.1. Next, we present how these difference matrices are integrated with physical conservation laws, as implicit constraints, to ensure global consistency in flow dynamics in Sec. 3.2. Finally, we propose a new message-passing equation that incorporates these constraints, demonstrating its capability to capture complex flow dynamics in Sec. 3.3.

### 3.1. Discretized Difference Matrices for Explicit Local Directionality Encoding

Discretized difference matrices encode directional sensitivity by approximating spatial gradients in discrete form, providing a framework for modeling local variations and directional dependencies in flow dynamics. Inspired by recent numerical methods (LeVeque, 2002), the discretized difference update process can be interpreted as a multi-layer GNN with specific adjacency matrices.

To see this, we start from its general format. A time-space discretized physics process can be described as:

$$\boldsymbol{\mu}^{t+1} = \boldsymbol{\mu}^t + \Delta t \frac{\partial \boldsymbol{\mu}^t}{\partial x}, \qquad (2)$$

where $\boldsymbol{\mu}^t \in \mathbb{R}^{|V|}$ is a row vector of nodal feature matrix $\mathbf{X}$, representing the state of a variable (*i.e.*, flux volume) in the graph $G$ at time $t$. $\Delta t$ is the time step that defines the

temporal resolution. $\partial \boldsymbol{\mu}^t / \partial x$ represents the spatial gradient of $\boldsymbol{\mu}$ along the $x$-direction (*i.e.,* the edge direction), which captures local directional variations. $\boldsymbol{\mu}^{t+1}$ is the updated state of this variable after incorporating temporal and spatial changes. Approximating the gradient $\partial \boldsymbol{\mu}^t / \partial x$ in Eq. (2) using a discretized difference scheme, we have:

$$\boldsymbol{\mu}^{t+1} = \boldsymbol{\mu}^t + \alpha \hat{D} \boldsymbol{\mu}^t = (I + \alpha \hat{D}) \boldsymbol{\mu}^t, \qquad (3)$$

where $\hat{D}$ is the discretized difference matrix, $I$ is the identity matrix, and $\alpha = \Delta t / \Delta x$ is a scalar balancing the time step $\Delta t$ and the spatial step $\Delta x$. Eq. (3) links the discrete update process to the graph adjacency operator $(I + \alpha \hat{D})$, encoding both the original topology and local variations.

To capture directionality in regions with rapid transitions and local changes, we leverage the the upwind scheme that allows for modeling directional dependencies in dynamic systems (Bermudez & Vazquez, 1994), further ensuring numerical stability in our framework. The upwind scheme approximates gradients as $\partial \mu / \partial x \approx (\mu_i - \mu_j) / \Delta x$, where $\mu_i$ and $\mu_j$ are the $i$-th and $j$-th entries of $\boldsymbol{\mu}^t$, representing the physical quantities of nodes $v_i$ and $v_j$ at time $t$, respectively. $\Delta x$ represents the spatial step between nodes $v_i$ and $v_j$, with $v_j$ being the upstream neighbor of $v_i$. This equation prioritizes upstream information, aligning with the physical reality of flows propagating downstream.

As such, we can construct the discretized difference matrix $\hat{D}$ based on the graph structure, where the nodes represent spatial locations and the edges encode directional dependencies. For a node $v_i$ and its upstream neighbor $v_j$, the $(i, j)$-th entry of $\hat{D}$ can be defined as:

$$\hat{D}_{ij} = \begin{cases} 1, & \text{if } i = j, \\ -1, & \text{if } j \text{ is the upstream node of } i, \\ 0, & \text{otherwise.} \end{cases}$$

In the first row of $\hat{D}$, we enforce directionality from $v_1$ to $v_0$, allowing $v_0$ to receive information without explicit initial conditions while preserving correct flow dependencies.

Using edge vector $\mathbf{e}_{ij}$, we define two enhanced difference matrices $D_1$ and $D_2$ as follows.

$$D_1 = \frac{1}{\Delta x} \hat{D}, \quad D_2 = \frac{\Delta z}{\Delta x} \hat{D}, \qquad (4)$$

where $\Delta x = \phi_1(\mathbf{e}_{ij})$ and $\Delta z = \phi_2(\mathbf{e}_{ij})$, and $\phi_1$ and $\phi_2$ are learnable mappings such as multi-layer perceptrons (MLPs).

The intuition behind Eq. (4) is that, $\Delta x$ represents the spatial distance, governing the propagation rate, while $\Delta z$ reflects elevation differences, encoding gravitational effects. These specific matrices arise naturally from the chosen PDEs, but the underlying approach of using difference operators derived from graph topology (e.g., spatial adjacency or functional relationships) is generalizable (Grady & Polimeni,

2010). Incorporating $D_1$ and $D_2$ into the GNN framework enhances its ability to model directional dependencies, aligning with natural flow dynamics while maintaining stability.

## 3.2. Incorporating Physical Equations as Implicit GNN Training Regularizers

Physical equations provide implicit constraints by enforcing global conservation laws, while difference matrices encode directional sensitivity at the local level. In fact, our tailored difference matrices can be applied to a wide range of flow models and are particularly suited for integration with physical equations. This allows for the incorporation of problem-specific constraints to address particular applications. In this study, we demonstrate that the difference matrices allow for the incorporation of problem-specific constraints to address two different applications.

CASE 1: S-V EQUATION FOR RIVER NETWORKS

In river flow modeling, the Saint-Venant (S-V) equations are widely used to describe water flow dynamics (Wu, 2007; Vreugdenhil, 2013). These equations establish the conservation of mass and momentum as the fundamental physical principles governing river water movement.

**Conservation of Momentum.** The momentum conservation equation accounts for the forces influencing water movement, including gravity and friction. $Q = h \cdot u$ represents discharge, where $h$ is water depth and $u$ is velocity. $g$ is gravitational acceleration, and $z(x)$ is bed elevation. The momentum conservation equation is given by:

$$\frac{\partial Q}{\partial t} + \frac{\partial}{\partial x}\left(\frac{Q^2}{h} + \frac{1}{2}gh^2\right) = -gh\frac{\partial z}{\partial x} - f, \quad (5)$$

where $f = gn^2 Q|Q|/h^{4/3}$ denotes the friction term, with $n$ being the Manning coefficient.

Eq. (5) inherently captures the directionality of river flow. The term $-gh\frac{\partial z}{\partial x}$ ensures downhill water movement by aligning with the steepest descent, while the inertial term $\frac{\partial}{\partial x}\left(\frac{Q^2}{h} + \frac{1}{2}gh^2\right)$ maintains consistency in flow dynamics. By integrating difference matrices with these terms, the solution space is constrained to adhere to fundamental physical laws while ensuring stability and directionality.

In implementation, we further simplify Eq. (5) by neglecting water depth and friction effects, which yields:

$$\frac{\partial \mathbf{u}}{\partial t} + \mathbf{u} \cdot \frac{\partial \mathbf{u}}{\partial x} = -g\frac{\partial z}{\partial x}, \quad (6)$$

where $\mathbf{u} \in \mathbb{R}^{|\mathcal{V}|}$ is the vector of fluid velocity of all nodes at time $t$, and $z$ denotes elevation.

**Discretization.** Eq. (6) can be discretized in both time and space to facilitate numerical implementation. Using dis-

cretized difference matrices and rearranging terms leads to the update rule for velocity at each node $i$:

$$u_i^{t+1} = u_i^t - \Delta t \left( u_i^t \frac{u_{i+1}^t - u_i^t}{\Delta x} + g\frac{z_{i+1} - z_i}{\Delta x} \right), \quad (7)$$

where $u_i^t$ is the scalar velocity at node $i$ and time step $t$, and $z_i$ is the scalar elevation at node $i$.

**Integration with Difference Matrices.** To enhance GNNs for modeling spatial variations and flow directions, we initially replace the adjacency matrix with a generalized difference matrix in Eq.(3). This general framework provides a foundation for directional sensitivity and spatial variation modeling in GNNs. To further align with physical principles, the generalized difference matrix Eq.(3) is adapted to the governing PDE by incorporating specific physical properties. For example, in the momentum equation, Eq.(3) is replaced with a PDE-specific difference matrix, which encodes elevation-based gradients and flow transport. The updated velocity at node $i$ is then computed as:

$$u_i^{t+1} = u_i^t - \alpha \left( u_i^t (\hat{D}u^t)_i + g(\hat{D}z)_i \right), \quad (8)$$

where $(\hat{D}u^t)_i$ represents velocity differences, and $(\hat{D}z)_i$ encodes elevation-driven effects. Parameter $\alpha$ controls the influence of the difference matrix in the overall update rule.

CASE 2: A-R EQUATION FOR TRAFFIC NETWORKS

In traffic flow modeling, the Aw-Rascle (A-R) equations are widely used to describe vehicle dynamics by extending classical traffic flow models. These equations provide a hyperbolic system of conservation laws to model traffic behavior. (Aw & Rascle, 2000).

**Conservation of Mass.** The mass conservation equation governs the evolution of vehicle density $\rho(x, t)$ over time and space. Representing $\rho$ as the vehicle density and $u(x, t)$ as the velocity, the conservation of mass is expressed as:

$$\frac{\partial \rho}{\partial t} + \frac{\partial(\rho u)}{\partial x} = 0. \quad (9)$$

This equation ensures that the total number of vehicles is conserved across the traffic network, where $\rho u$ represents the traffic flux. The coupling of vehicle density $\rho(x, t)$ and velocity $u(x, t)$ in $\rho u$ captures the effects of local density variations and their influence on traffic movement. This formulation allows the AR model to effectively represent traffic dynamics in real-world scenarios.

**Discretization.** The mass conservation equation for traffic networks can be discretized in both time and space to facilitate numerical implementation. Using discretized difference schemes and rearranging terms, we derive the update rule

for density at each node:

$$\rho_i^{t+1} = \rho_i^t - \Delta t \left( u_i^t \frac{\rho_{i+1}^t - \rho_i^t}{\Delta x} + \rho_i^t \frac{u_{i+1}^t - u_i^t}{\Delta x} \right), \quad (10)$$

where $\rho_i^t$ is the scalar density at node $i$ and time step $t$, and $u_i^t$ is the scalar velocity at node $i$. Eq. (10) accounts for both the spatial variation of density and the effect of velocity gradients, ensuring consistency with total traffic mass.

**Integration with Difference Matrices.** The updated traffic density at node $i$ is then computed as:

$$\rho_i^{t+1} = \rho_i^t - \alpha \left( u_i^t (\hat{D}\rho^t)_i + \rho_i^t (\hat{D}u^t)_i \right), \quad (11)$$

where $(\hat{D}u^t)_i$ represents velocity differences, and $(\hat{D}\rho^t)_i$ encodes density-driven effects. $\alpha$ is a balancing factor. These terms approximate the spatial derivatives of velocity and density, respectively, using a difference operator $\hat{D}$.

### 3.3. Unifying Difference Matrices and PDEs in Message-Passing Layers

We integrate physical knowledge into the GNN training process by unifying difference matrices and PDEs to enhance modeling for flood and traffic flow dynamics.

CASE 1: MESSAGE-PASSING FOR FLOOD PREDICTION

We extrapolate the message-passing function indicated in Eq. (1) by explicitly distinguishing the roles of $\hat{D}$ in capturing both local gradients and elevation-driven effects, based on Eq. (8). Specifically, we follow Eq. (4) to decompose $\hat{D}$ into $D_1$ and $D_2$. The message-passing process in our `PhyNFP` framework for river network can be formulated as:

$$\mathbf{h}^{l+1} = \mathbf{h}^l - \Delta t \left( \mathbf{h}^l \odot (D_1 \mathbf{h}^l W_1) + \hat{g} \cdot (D_2 \mathbf{h}^l W_2) \right), \quad (12)$$

where $\mathbf{h}^l \in \mathbb{R}^{|V| \times d}$ represents the node embedding matrix at layer $l$, initialized as $\mathbf{h}^0 = \mathbf{X} \in \mathbb{R}^{|V| \times (t \cdot p)}$. As deeper message-passing layers enables information exchange between a node and its topologically more faraway neighbors, simulating longer-term system dynamics, we use the update from $l$ to the $(l+1)$-th layer to surrogate the accumulation of changes over two consecutive time steps in PDEs. The difference matrices $D_1 \in \mathbb{R}^{|V| \times |V|}$ captures local spatial gradients and $D_2 \in \mathbb{R}^{|V| \times |V|}$ incorporates elevation-driven dynamics influenced by graph topology. $W_1$ and $W_2 \in \mathbb{R}^{d \times d}$ are learnable parameters that learn node embeddings within the same dimension, allowing for the element-wise multiplication $\odot$. $\Delta t$ and $\hat{g}$ are learnable scalars that modulating the influence of spatial and elevation-driven terms and scales the contribution of elevation-driven dynamics, respectively.

In our tailored message-passing Eq. (12), the term $D_1 \mathbf{h}^l$ captures local spatial derivatives, reinforcing directional

information, and $D_2 \mathbf{h}^l$ integrates elevation variations that influence flow propagation. The learnable weights $W_1$ and $W_2$ further refine these representations, ensuring consistency across layers. Using learnable $\Delta t$ and $g$ allows for additional flexibility, making GNNs adaptive to based on training data while preserving the underlying physical principles. Leveraging Eq. (12), our `PhyNFP` empowers GNNs to model rapid spatial and directional variations, improving performance in predicting flux volumes of river networks.

CASE 2: MESSAGE-PASSING FOR TRAFFIC FLOW

To enhance directional sensitivity in traffic networks, we reformulate the traffic flow conservation Eq. (9) by regulating the contributions of traffic density and velocity variations:

$$\mathbf{h}^{l+1} = \mathbf{h}^l - \Delta t \left( \mathbf{h}^l \odot (D_1 \mathbf{v}^l W_1) + \mathbf{v}^l \odot (D_1 \mathbf{h}^l W_2) \right), \quad (13)$$

where the node embedding matrix at layer $l$ remains $\mathbf{h}^l \in \mathbb{R}^{|V| \times p}$, but initialized as $\mathbf{h}^0 = \text{MLP}_h(\mathbf{X}) \in \mathbb{R}^{|V| \times d}$. Denoted by $\mathbf{v}^l \in \mathbb{R}^{|V| \times p}$ an embedding matrix, initialized as $\mathbf{v}^0 = \text{MLP}_v(\mathbf{X}) \in \mathbb{R}^{|V| \times d}$ that extracts velocity property from raw nodal features. Here, we only use $D_1 \in \mathbb{R}^{|V| \times |V|}$ that encodes spatial variations in both traffic density and velocity, reflecting how traffic propagates through the network. $W_1$ and $W_2 \in \mathbb{R}^{d \times d}$ are learnable weights, and $\Delta t$ is a learnable scalar used to balance local traffic variations and temporal propagation. In the message-passing Eq. (13) tailored for traffic network, $D_1 \mathbf{v}^l$ captures velocity gradients that drive traffic movement, while $D_1 \mathbf{h}^l$ accounts for density variations that influence traffic congestion. Therefore, although both terms share the same difference matrix $D_1$, their physical interpretations differ. Namely, $D_1 \mathbf{v}^l$ determines velocity-induced flow adjustments, and $D_1 \mathbf{h}^l$ regulates density-based congestion propagation. The learnable matrices $W_1$ and $W_2$ refine these interactions, enabling GNNs to adapt to free-flow and congested conditions. By making $\Delta t$ learnable, GNNs can adjust their sensitivity to real-time traffic conditions, providing a physics-aware approach to traffic prediction.

## 4. Experiments

**Datasets.** Two datasets collected from real-world directed graphs are used. 1) _River_, preprocessed from LamaH-CE2 (Klingler et al., 2021), which documents historical discharge and meteorological measurements with hourly resolution in the Danube river network. It consists of 358 nodes and 357 edges. Five nodal features include discharge, surface pressure, precipitation, temperature, and soil moisture. Three edge features include length, slope, and distance. 2) _Traffic_, preprocessed from PEMS-04 (Yu et al., 2018), that comprises traffic flow records collected from roadside sensor stations. It consists of 307 nodes and 340 edges. Three nodal features include flow, occupy and speed. Edge fea-

*Table 1.* Comparative results in two datasets, with prediction horizon $n = 6$ and MSE measured on the volume rescaled by normal score.

| Datasets | River Network | | | | Traffic Network | | | |
|---|---|---|---|---|---|---|---|---|
| | Forward (F) | Reverse (R) | $DS$ | $RDS$ | Forward (F) | Reverse (R) | $DS$ | $RDS$ |
| PhyNFP (Ours) | 0.0801 | 0.0906 | +0.0105 | - | 0.0696 | 0.0724 | +0.0028 | - |
| PhyNFP$_{DM}$ (ablation) | 0.0898 | 0.0961 | +0.0063 | -40.0% | 0.0721 | 0.0738 | +0.0017 | -39.3% |
| GWN | 0.1101 | 0.1132 | +0.0031 | -70.5% | 0.0709 | 0.0706 | -0.0003 | -110.7% |
| MP PDE Solver | 0.1126 | 0.1082 | -0.0044 | -141.9% | 0.0700 | 0.0711 | +0.0011 | -60.7% |
| MPNN | 0.1170 | 0.1182 | +0.0012 | -88.6% | 0.0713 | 0.0720 | +0.0007 | -75.0% |
| GraphSAGE | 0.1224 | 0.1149 | -0.0075 | -171.4% | 0.0724 | 0.0712 | -0.0012 | -142.9% |
| GAT | 0.1233 | 0.1265 | +0.0032 | -69.5% | 0.0768 | 0.0776 | +0.0008 | -71.4% |
| GNO | 0.1247 | 0.1265 | +0.0018 | -82.9% | 0.0757 | 0.0765 | +0.0008 | -71.4% |
| GCN | 0.1365 | 0.1357 | -0.0008 | -107.6% | 0.0769 | 0.0778 | +0.0009 | -67.9% |

ture is the distance between nodes. Input features over $W$ hours are concatenated along the feature dimension before being fed into the models. The ground-true flux volumes $\mathbf{Y} \in \mathbb{R}^{|V|}$ are available for all nodes in both datasets. We normalize all physical variables including the nodal features and output volume to the same scale in an element-wise fashion using standard score (LeCun et al., 2002).

**Metrics.** Following the prior art (Kirschstein & Sun, 2024), we benchmark the models in the regime of supervised node regression. Given a certain amount of $W$ (*i.e.,* a window size) observations of flux volume of all nodes, our task is to predict the volume $n$ hours ahead, namely, the prediction horizon is $n$. We set $W = 24$ for training and $n = 6$ for the lead time prediction for applicability. The prediction discrepancy is gauged by the mean squared error (MSE) averaged over all nodes, namely, $\ell(\hat{\mathbf{Y}}, \mathbf{Y}) = (1/|V|)\|\mathbf{Y} - \mathbf{Y}\|_2^2$.

**Direction Sensitivity.** To substantiate the effectiveness of distinguishing edge directions, we benchmark the experiments in the original graph datasets (denoted as *Forward*) and their inverse counterparts, where the direction on every edge is reversed (denoted as *Reverse*). We define direction sensitivity of a certain model $M$ as $DS(M) = \ell_M(\text{Reverse}) - \ell_M(\text{Forward})$, where $\ell_M$ indicates the MSE loss of $M$, and intuitively its performance in the *Forward* setting should excel. Further, we can define the relative direction sensitivity as $RDS(M_1, M_2) = (DS(M_2) - DS(M_1))/DS(M_1)$ between two models $M_1$ and $M_2$.

**Competitors.** Eight models are identified for comparative study, divided into three categories as follows. First, the traditional GNNs including *1)* Graph Convolutional Network (GCN) (Wu et al., 2019) that propagates node features in spectral domain, *2)* Graph Attention Network (GAT) (Veličković et al., 2018) that furthers GCN with attention mechanism, *3)* Message-Passing Neural Network (MPNN) (Gilmer et al., 2020) that employs general feature aggregation and update functions, *4)* GraphSAGE (Hamilton et al., 2017) for inductive representation learning, and

*5)* Graph Wavelet Network (GWN) (Xu et al., 2019) that uses wavelet transforms to capture high-frequency components. Compared with those traditional GNNs, the efficacy of our PhyNFP performs in preserving directional sensitivity, capturing high-frequency components, and improving flux predictive performance.

Second, the graph learning models for problem-solving in physical systems. They include *6)* Message-Passing PDE Solver (MP-PDE Solver) (Brandstetter et al., 2022) that uses message passing to approximate PDE solutions, capturing spatial and temporal dynamics without enforcing physical constraints, and *7)* Graph Neural Operator (GNO) (Li et al., 2020) that learns mappings between function spaces, so to adapt to spatial and temporal dynamics without enforcing physical constraints. Both MP-PDE Solver and GNO are data-driven approaches that do not explicitly incorporate physical laws. Comparing with them help evaluating how well the proposed PhyNFP balances physical consistency and data-driven modeling.

Third, for ablation study, we propose a variant reduced from our proposed approach: *8)* PhyNFP$_{DM}$, which only uses the basic adjacency information constructed from discretized difference matrices in Eq. (3) for message-passing. This variant does not incorporate PDEs into its GNN training. A comparison with it will demonstrate the effectiveness of incorporating specific PDEs as constraints for domain problems as specified in Sec. 3.2.

**Results and findings.** Table 1 presents the MSE and directional sensitivity scores (DS and RDS) for different models on river and traffic networks. We answer the following research questions (RQs) based on the results.

**RQ1** *How does the proposed PhyNFP framework improve flux prediction over the compared graph learners?*

Our method achieves the best overall performance in both datasets. To quantify these improvements, we compute

| Method | Forward (F) | | |
|--------|-------|-------|-------|
| $n$ | 3 | 6 | 9 |
| PhyNFP | 0.0514 | 0.0801 | 0.1087 |
| GAT | 0.0617 | 0.1233 | 0.1433 |
| GCN | 0.0632 | 0.1365 | 0.1481 |

*Table 2.* Comparative results with baseline GNN models in river network with varying prediction time horizon $n$.

the average Forward MSE, DS, and RDS across all baseline models, including the ablation version of our method, and compare them with our approach. Specifically, in the river network, the compared methods on average achieve an MSE in the Forward setting as 0.1170, whereas our method achieves 0.0801, representing a 31.6% reduction in the MSE prediction error. The DS score of PhyNFP is 0.0105, outperforming the compared methods that on average arrive at 0.0004, leading to $26\times$ improvement in relative directional sensitivity (RDS). In the traffic network, the baseline average Forward MSE is 0.0732, while our method achieves 0.0696, reducing error by 4.9%. The baseline DS is 0.0006, while our model achieves 0.0028, corresponding to a $3.6\times$ improvement in RDS. These results indicate that traditional graph-based models, including GNNs and graph-aware PDE solvers, struggle with directional sensitivity, while our method significantly enhances topology-aware modeling, resulting in improvement in flux prediction.

**RQ2** *Does domain-specific physics information helpful in graph-related flux prediction tasks?*

To understand performance variations, we categorize models into two groups: physics-guided models (*e.g.*, MP-PDE Solver, GNO and PhyNFP$_{DM}$ (ablation)) and purely data-driven models (*e.g.*, GCN, GraphSAGE, GWN, GAT, and MPNN). In the river network, physics-guided models have an average Forward MSE of 0.1090, which is 26.5% higher than our 0.0801, while their average DS is 0.0012 compared to our 0.0105, resulting in a 88.3% lower RDS. Purely data-driven models perform similarly, with an average Forward MSE of 0.1218 (34.3% higher than ours) and a DS of -0.0002, leading to a 101.9% lower RDS. In the traffic network, physics-guided models have an average Forward MSE of 0.0726, which is 4.1% higher than our 0.0696, and an average DS of 0.0012, making their RDS 57.1% lower. Purely data-driven models show an average Forward MSE of 0.0736 (5.4% higher than ours) and a DS of 0.0002, leading to a 92.8% lower RDS. Physics-guided models achieve lower Forward MSE and better DS/RDS than purely data-driven models, showing that incorporating physical knowledge helps with directional flow modeling. However, those physics-guided models perform worse than our method.

We observe weaker directional sensitivity (DS/RDS) in the

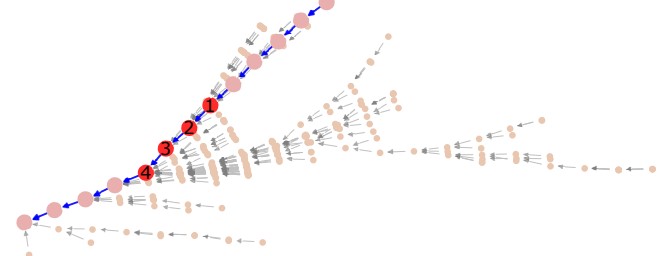

(a) Graph topology of the River dataset.

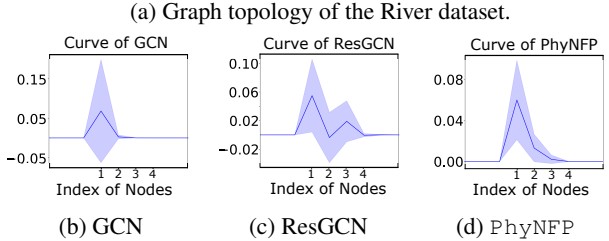

| (b) GCN | (c) ResGCN | (d) PhyNFP |

*Figure 2.* Trends of prediction results in response to a local and rapid flux change. (a) The change occurs in node $v_1$ and propagates to the downstream nodes $v_2$, $v_3$, and $v_4$. The responsive prediction errors (in MSE) across the four nodes from (b) GCN, (c) ResGCN, and (d) our PhyNFP framework.

Traffic network compared to the River network, stemming from two main factors. First, the governing physics differ: river flow modeling (momentum-based) enforces direction more strongly than traffic flow modeling (mass/density-based). Second, their graph structures contrast: the River network is largely tree-like, inherently supporting directed information flow during message passing. The Traffic network, however, contains many cycles, which allow message passing routes that can counteract strict directionality, thus blurring the distinction between the deliberately set forward and reverse topologies. Both the physics and the cyclic structure therefore make achieving high directional sensitivity more challenging in the traffic network.

**RQ3** *What is the impact of time horizon in prediction?*

Table 2 presents the MSE for different methods under forward flow in the river network as the prediction horizon $n$ increases. We observe that, as $n$ increases from 3 to 9, all models show increasing errors, reflecting the challenge of long-horizon predictions. However, the error growth is significantly slower for our method, increasing by only 0.0573 (from 0.0514 to 0.1087), whereas GAT and GCN experience larger increases of 0.0816 and 0.0849, respectively. Meanwhile, our method consistently achieves lower MSE across all horizons, demonstrating that our method maintains better stability and robustness over longer horizons.

**RQ4** *How well is our proposed PhyNFP in capturing high-frequency components and local and rapid changes?*

Figure 2 shows the topology of the river dataset and the prediction results under perturbations (*i.e.,* simulating high-frequency components). The y-axis represents the difference between perturbed and unperturbed predictions. In Figure 2(b), (c), and (d), the blue line indicates the mean prediction over the test set, while the gray area represents the $3\sigma$ confidence interval.

In Figure 2(a), the topology of the river network is depicted as a tree-like structure. A perturbation ($+0.5$) on $v_1$ is introduced at the final time step to observe its effect on downstream nodes. This perturbation amount is considerable given the data has been normalized. Figure 2(b) shows that GCN fails to propagate the perturbation to downstream nodes, indicating that GCN struggles to capture high-frequency components in nodal representations. Figure 2(c) illustrates that, although ResGCN demonstrates propagation at certain extend, it introduces more errors. For example, the perturbation at $v_2$ should increase the value; instead, ResGCN causes a decrease, showing that it lacks consistency in flow modeling. In Figure 2(d), `PhyNFP` successfully propagates the perturbation to multiple downstream nodes without introducing error responses. This demonstrates that our model effectively captures upstream-to-downstream dependencies while maintaining physical consistency.

**RQ5** *How well is `PhyNFP` able to extract physics underlying PDE in solving the inverse problem?*

To validate that `PhyNFP` truly incorporates the physical dynamics described by the PDEs, and to confirm our hypothesis about the reverse task behaving as an ill-posed inverse problem, we analyze its behavior in the reverse setting, particularly when subject to local perturbations. As established in Section 2, solving hyperbolic PDEs upstream in space is inherently unstable and sensitive to high-frequency perturbations. A model that correctly captures these physics should exhibit signs of this instability in the reverse setting, unlike standard GNNs which tend to smooth out such effects.

Figure 4 in the Appendix illustrates the model responses to a local perturbation injected at a node ($v_1$) in the reverse river network setting. For `PhyNFP`, the perturbation incorrectly propagates upstream (to $v_2, v_3, \dots$). While physically incorrect for forward flow, this behavior is the expected signature of solving the PDE backward from downstream data. The response of `PhyNFP` in this setting, which directly reflects the ill-posed and potentially unstable nature of this inverse problem, thus demonstrates its capture of the PDE-encoded dynamics. In contrast, GCN and ResGCN show minimal upstream response, suppressing the perturbation due to their low-pass filtering property, which highlights their insensitivity to such physical dynamics and direction reversal.

Further evidence comes from the learned time step parameter $\Delta t$. As shown in Figure 3, $\Delta t$ stabilizes at a higher

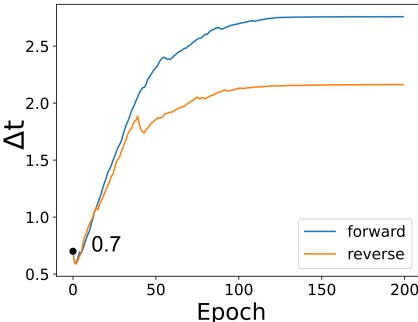

*Figure 3.* Evolution of the learned time-step parameter $\Delta t$ over training epochs for the forward and reverse settings in the river network. The model starts with an initial $\Delta t = 0.7$.

value in the forward setting, whereas in the reverse setting it converges to a value approximately 21.58% smaller. This reflects the need for tighter step sizes to ensure stability when solving ill-posed inverse problems (Baumeister, 1987).

These results demonstrate that `PhyNFP` effectively extracts the physics embedded in the governing PDE. Its response to upstream perturbations and the adaptive adjustment of the learned time step $\Delta t$ reflect its ability to capture the instability associated with solving the PDE in reverse.

## 5. Related Work

We identify three thrusts of related studies as follows.

**Physics-based Flood Forecasting**   Traditional hydrodynamic models, based on the Saint-Venant equations, are widely used for flood forecasting due to their detailed physical representation of river flows. These models solve PDEs to simulate key hydrological variables such as water flow, velocity, and depth across spatial grids. Examples include HEC-RAS (Hydrologic Engineering Center's River Analysis System)(Brunner, 2002), HL-RDHM (Hydrology Laboratory Research Distributed Hydrologic Model)(Moreda et al., 2006; Fares et al., 2014), and SWAT (Soil and Water Assessment Tool) (RS & Williams, 1998), which approximate water movement based on river topology, rainfall intensity, and terrain features. Despite their accuracy, these models demand extensive computational resources due to fine-grained spatial and temporal discretization, making real-time adaptation challenging.

**Physics-based Traffic Flow Prediction.**   Traditional traffic flow models are formulated as partial differential equations (PDEs) to capture the macroscopic dynamics of vehicle movements. Classical models such as the Lighthill-Whitham-Richards (LWR) model(Leclercq, 2007) describe traffic density evolution using conservation laws, while the Aw-Rascle-Zhang (ARZ) model(Aw & Rascle, 2000; Yu

& Krstic, 2019) extends LWR by incorporating velocity-dependent pressure terms to model traffic congestion more accurately. Additionally, the second-order macroscopic models, such as the Payne-Whitham model (Jin & Zhang, 2003), introduce momentum conservation to capture driver reaction behaviors. These models provide interpretable theoretical frameworks but require detailed parameter calibration and struggle to adapt to dynamic traffic conditions. Furthermore, alternative data-driven approaches, such as multi-stream fuzzy learning or topology-based fuzzy networks, aim to address uncertainty and dynamic changes in transportation systems (Yu et al., 2020; 2022).

**Physics-Informed/Guided Graph Neural Networks.** The integration of physics with GNNs has proven effective for solving systems governed by PDEs. Graph Neural Operators (GNOs) (Li et al., 2020) use graph kernels to learn mappings between function spaces, enabling efficient PDE solutions across varying domains. Graph Neural Diffusion (GRAND) (Chamberlain et al., 2021) models diffusion processes on graphs, capturing long-range dependencies and incorporating physical principles. Graph Neural Ordinary Differential Equations (GDEs) (Poli et al., 2019) describe node feature evolution as continuous trajectories governed by ODEs, offering adaptive computation for dynamic processes. Message Passing Neural PDE Solvers (Brandstetter et al., 2022) leverage graph structures to propagate information and approximate PDE solutions. These approaches illustrate the synergy between physics-based modeling and GNNs in scientific and engineering tasks. In addition, PDE-Net(Long et al., 2018) embed differential operators into neural networks, enhancing interpretability and enforcing physical constraints. Some studies further demonstrate how to leverage difference matrices to encode physical laws (Liu et al., 2024). Spatio-temporal graph neural networks (ST-GNNs) have been shown to enhance predictions by integrating rainfall-runoff data with river topologies in complex networks for flood forecasting (Roudbari et al., 2024; Kazadi et al., 2023; Farahmand et al., 2023) and traffic flow modeling (Bui et al., 2022; Guo et al., 2019). These approaches enable scalable and consistent solutions for tasks like flood prediction and urban traffic forecasting.

However, deep GNNs, including physics-informed ones, often encounter the over-smoothing problem, where node features tend to become overly similar with increasing network depth. This limitation restricts their ability to capture high-frequency components in flow dynamics. Some studies have attempted to mitigate the over-smoothing with PDE. For example, PDE-GCN constructs GCNs by discretizing hyperbolic PDEs (Eliasof et al., 2021). Rusch et al. introduce GraphCON, a framework based on coupled oscillators (Rusch et al., 2022), and further propose Gradient Gating ($G^2$) to control information flow and address oversmoothing

(Rusch et al., 2023). Our proposed `PhyNFP` framework is also grounded in PDEs. Its ability to operate effectively with up to 19 layers as in (Kirschstein & Sun, 2024), in contrast to standard GNNs, stems from the use of upwind schemes in the difference matrices and the enforcement of physical consistency. These mechanisms inherently stabilize the message-passing process without relying on explicit over-smoothing regularization.

## 6. Conclusion

This paper explored the limitation of GNNs in modeling directed graph-based flow systems, where physical dynamics are governed by directional dependencies. Our analysis demonstrates that GNNs often exhibit directional insensitivity due to their inherent low-pass filtering effect during message passing. This limitation prevents them from effectively capturing high-frequency variations in flow dynamics, such as abrupt flux changes and sharp transitions. As such, standard GNNs struggle with inverse problems, where accurate representation of directional and localized changes is crucial. In response, we proposed the `PhyNFP` framework, which integrates physical principles into GNN training to enhance directional sensitivity and improve performance in flow dynamics modeling. `PhyNFP` consists of two main components, namely 1) discretized difference matrices that encode directional gradients and local variations, preserving high-frequency information that traditional adjacency-based GNNs filter out, and 2) physical law regularization, whereby incorporating global physical equations such as momentum conservation into the training process, `PhyNFP` ensures the compliance between predictive results and the underlying physics. Extensive experiments on real-world river and traffic networks demonstrate that `PhyNFP` can better capture both high-frequency and directional dependencies, leading to significant improvements over baseline GNN models and graph-aware PDE solvers in terms of prediction accuracy and flow representation, substantiating the effectiveness and promising modeling of integrating domain-specific physical knowledge into graph learning regimes.

In future work, we aim to extend our framework to incorporate boundary and initial conditions of the governing PDEs, which lend a more faithful representation of fluid dynamics. Their explicit integration may further improve the physical fidelity and predictive accuracy of our model.

## Acknowledgement

This work has been supported in part by the National Science Foundation (NSF) under Grant Nos. IIS-2441449, IIS-2236578, IOS-2446522, IIS-2236579, IIS-2302786, IOS-2430224, the Science Center for Marine Fisheries (SCeM-FiS), and the Commonwealth Cyber Initiative (CCI).

## Impact Statement

This research contributes to the advancement of Machine Learning. Its application to modeling flow-based systems can yield significant societal benefits, particularly in environmental resource management and transportation optimization. Our method can improve predictions for floods, droughts, pollution, and traffic congestion, thereby aiding sustainable development and disaster preparedness. By integrating physical laws, our approach also fosters trust in AI, mitigating overreliance on purely data-driven models. This work may spur innovation in diverse fields like healthcare, energy, and climate science, and encourage interdisciplinary collaboration. We identify no specific ethical implications requiring special highlighting.

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

**Overview.** This supplementary material provides a results figure in reverse setting (in Appendix A) and additional analysis to support our main paper in two key aspects. First, we evaluate the effectiveness of the proposed discretized difference matrices using the Discrete-Time Fourier Transform (DTFT), demonstrating that these matrices enhance the model sensitivity to high-frequency components, as detailed in Appendix B. Second, we formulate the flux prediction problem on a directed graph with reversed edge directions as an inverse problem and provide a detailed rationale for this approach, which is discussed in Appendix C.

## A. Results under Perturbations in Reverse Setting

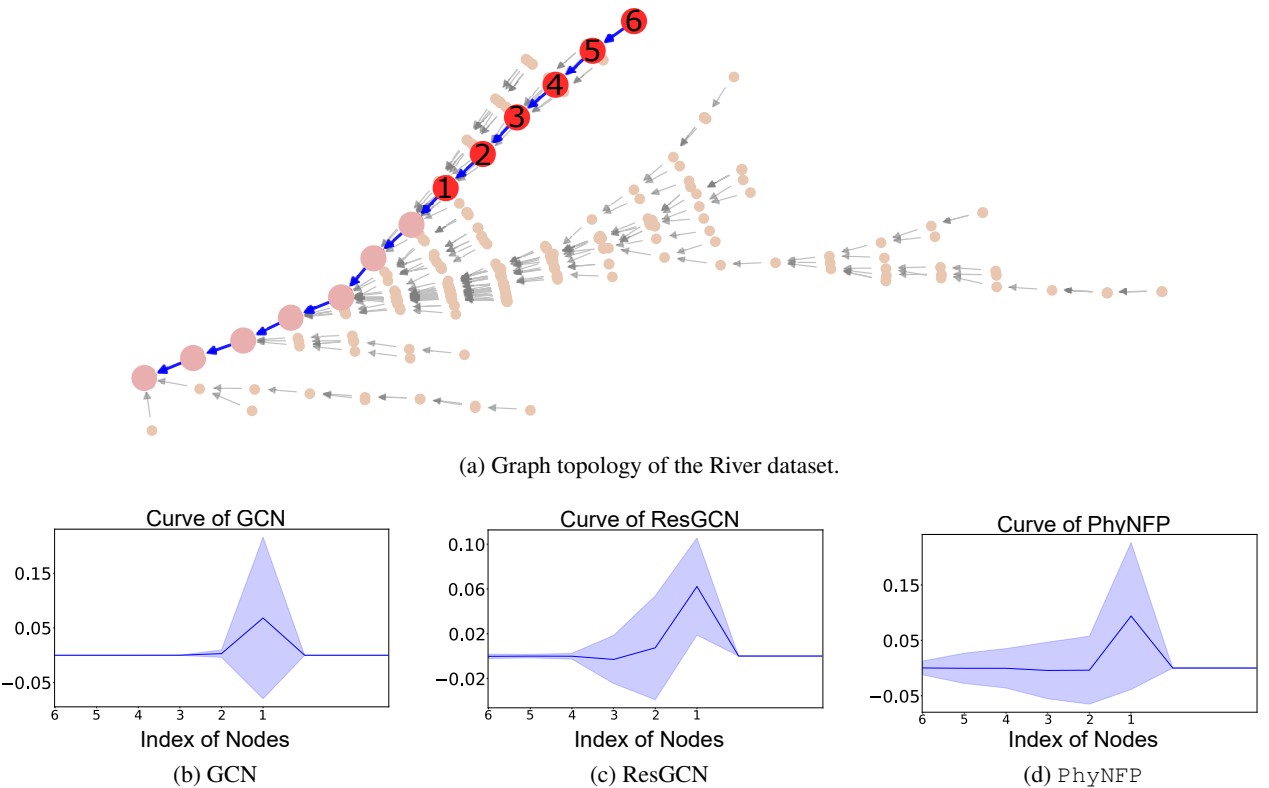

(a) Graph topology of the River dataset.

|  (b) GCN | (c) ResGCN | (d) `PhyNFP` |

*Figure 4.* Trends of prediction results in response to a local and rapid flux change. (a) The change occurs in node $v_1$ and propagates to the upstream nodes $v_2$ through $v_6$. The responsive prediction errors (in MSE) across the six nodes are shown for (b) GCN, (c) ResGCN, and (d) our `PhyNFP` framework.

## B. Difference Matrix and High-Frequency Sensitivity

### B.1. Definition of the Difference Matrix

In signal processing, the difference operator is used to capture variations in a signal. For a 1D sequential signal, the forward difference matrix $D$ of size $n \times n$ is defined as

$$
D = \begin{bmatrix}
1 & 0 & 0 & \cdots & 0 \\
-1 & 1 & 0 & \cdots & 0 \\
0 & -1 & 1 & \cdots & 0 \\
\vdots & \vdots & \ddots & \ddots & \vdots \\
0 & 0 & \cdots & -1 & 1
\end{bmatrix}.
$$

Applying $D$ to a discrete-time signal $\mathbf{x} = [x_1, x_2, \ldots, x_n]^T$ gives

$$D\,\mathbf{x} = \begin{bmatrix} x_1 \\ x_2 - x_1 \\ x_3 - x_2 \\ \vdots \\ x_n - x_{n-1} \end{bmatrix},$$

which captures the differences between consecutive elements, making it sensitive to rapid changes in the signal.

## B.2. DTFT of the Difference Matrix

To analyze the effect of $D$ in the frequency domain, we use the Discrete-Time Fourier Transform (DTFT). The DTFT of a discrete-time signal $x(n)$ is

$$X(e^{j\omega}) = \sum_{n=-\infty}^{\infty} x(n)\, e^{-j\omega n}.$$

Consider the difference equation

$$y(n) = x(n) - x(n-1).$$

Taking its DTFT:

$$Y(e^{j\omega}) = \sum_{n=-\infty}^{\infty} \big[x(n) - x(n-1)\big] e^{-j\omega n}.$$

Using the time-shift property $\mathcal{F}[x(n-k)] = e^{-j\omega k}\, X(e^{j\omega})$, we have

$$\mathcal{F}[x(n-1)] = e^{-j\omega}\, X(e^{j\omega}).$$

Thus,

$$Y(e^{j\omega}) = X(e^{j\omega}) - e^{-j\omega}\, X(e^{j\omega}) = \left(1 - e^{-j\omega}\right) X(e^{j\omega}).$$

Hence, the frequency response of the difference operator is

$$D(e^{j\omega}) = 1 - e^{-j\omega}.$$

## B.3. Magnitude Response of the Difference Operator

Writing $e^{-j\omega} = \cos\omega - j\sin\omega$,

$$D(e^{j\omega}) = (1 - \cos\omega) + j\sin\omega.$$

Its magnitude is

$$|D(e^{j\omega})| = \sqrt{(1 - \cos\omega)^2 + \sin^2\omega}.$$

Using $1 - \cos\omega = 2\sin^2(\omega/2)$, we get

$$|D(e^{j\omega})| = 2\left|\sin(\omega/2)\right|.$$

For $\omega \to 0$, $|D(e^{j\omega})| \to 0$, so low-frequency components are suppressed. For $\omega \to \pi$, $|D(e^{j\pi})| = 2$, so high-frequency components are amplified. Thus, $D$ acts like a high-pass filter.

## B.4. Composite Operator $I + \alpha D$

Since $I$ is the identity operator (preserving all frequencies) and $D$ is a high-pass filter, their combination

$$H_{\text{combined}} = I + \alpha\, D$$

balances the global structure ($I$) with local variations ($D$).

B.4.1. FREQUENCY RESPONSE OF $I + \alpha D$

Taking the DTFT of $H_{\text{combined}}$:

$$H_{\text{combined}}(e^{j\omega}) = 1 + \alpha(1 - e^{-j\omega}) = 1 + \alpha - \alpha\, e^{-j\omega}.$$

Using $e^{-j\omega} = \cos\omega - j\sin\omega$,

$$H_{\text{combined}}(e^{j\omega}) = (1 + \alpha) - \alpha\,\cos\omega \; + \; j\,\alpha\,\sin\omega.$$

B.4.2. MAGNITUDE RESPONSE

The magnitude is

$$|H_{\text{combined}}(e^{j\omega})| = \sqrt{(1 + \alpha - \alpha\cos\omega)^2 + (\alpha\sin\omega)^2}.$$

B.4.3. SPECIAL CASES

For $\omega = 0$:

$$|H_{\text{combined}}(e^{j0})|^2 = 1,$$

so low-frequency components are unchanged.

For $\omega = \pi$:

$$|H_{\text{combined}}(e^{j\pi})|^2 = (1 + 2\alpha)^2,$$

hence

$$|H_{\text{combined}}(e^{j\pi})| = |1 + 2\alpha|,$$

allowing control of high-frequency amplification by adjusting $\alpha$.

**Remark.** The operator $I + \alpha\, D$ can be tuned to preserve smooth trends while selectively enhancing or reducing sharp transitions, making it highly adaptable in various discrete signal and graph processing tasks.

## C. Hyperbolic PDEs and Reverse Characteristic Tracing

### C.1. General Form of Hyperbolic PDEs

A hyperbolic partial differential equation can often be written as

$$\frac{\partial u}{\partial t} + \frac{\partial f(u)}{\partial x} = 0, \tag{14}$$

where $u(x, t)$ depends on time $t$ and space $x$, and $f(u)$ is the flux function. If $f(u) = c\,u$ with a positive constant $c$, then

$$u_t + c\,u_x = 0,$$

indicating that information propagates at speed $c$. If $f(u) = \frac{u^2}{2}$, then

$$u_t + u\,u_x = 0,$$

where $f'(u) = u$ depends on the solution itself, leading to wave speeds that can vary in space and time.

### C.2. Forward and Reverse Characteristic Tracing

C.2.1. CHARACTERISTIC EQUATIONS AND FORWARD TRACING

From (14), one derives the characteristic form:

$$\frac{d}{dt}u(x(t), t) = u_t + \frac{dx}{dt}\,u_x = u_t + f'(u)\,u_x = 0. \tag{15}$$

This implies

$$\frac{du}{dt} = 0 \quad \implies \quad u\big(x(t), t\big) = \text{constant},$$

and

$$\frac{dx}{dt} = f'(u).$$

In the linear case $f'(u) = c$, characteristics are straight lines $x(t) = x_0 + c\,t$. If $f'(u)$ depends on $u$, different characteristics may cross or diverge. For forward tracing, the solution evolves from an initial condition at $t = 0$ along these characteristic lines.

### C.2.2. REVERSE TRACING AND FLOW DIRECTION REVERSAL

To reconstruct the state at $t = 0$ from known data at $t = T$, one must trace characteristics backward. In the simpler linear case $u_t + c\,u_x = 0$, suppose

$$u(x, t) = \int_{-\infty}^{\infty} \hat{u}(\omega, t)\, e^{\,i\,\omega\,x}\, d\omega,$$

then

$$\hat{u}(\omega, t) = \hat{u}(\omega, 0)\, e^{-\,i\,c\,\omega\,t}.$$

If only noisy observations $\hat{u}_{\text{obs}}(\omega, T)$ are available at $t = T$, the inverse solution at $t = 0$ retains high-frequency noise, often leading to large oscillations in the physical domain.

In a river network or directed graph, reversing edges from downstream to upstream has an analogous meaning: instead of following the natural (forward) downstream flow, one essentially attempts to trace information upstream. From a PDE perspective, this parallels reversing the direction of characteristics. While valuable for estimating upstream fluxes or initial states, such a reversed approach can suffer from noise amplification and multivalued solutions when no dissipation is present.

### C.3. Effects of Nonlinearity and Multivalued Solutions

When $f'(u)$ depends on $u$, characteristic speeds vary with the solution. Different characteristic curves may converge (forming shocks) or diverge (forming rarefactions), sometimes creating multiple values of the solution in the same region. Nonlinearity also causes spectral broadening, so different frequency components can interact and generate new high-frequency terms. Consequently, reverse reconstruction is more sensitive to noise and can become numerically unstable.

### C.4. Regularization and Stability

Typical techniques to stabilize reverse problems include:

1. Adding a small viscous term,

$$u_t + f(u)_x = \nu\, u_{xx}, \quad \nu > 0,$$

   to provide smoothing and suppress high-frequency oscillations.

2. Introducing constraints or penalties in the inverse problem,

$$\min_u \|\mathcal{A}(u) - b\|^2 + \lambda \|u\|^2, \quad \lambda > 0,$$

   to tame large oscillations in the reconstructed solution.

3. Applying smoothing to boundary or initial data to mitigate discontinuities and avoid severe multivalued paths.

**Remark.** Reversing edges from downstream to upstream in a graph to predict flux is essentially a reverse characteristic approach akin to hyperbolic PDE theory. While it enables upstream inference, it also highlights the need for regularization or dissipative mechanisms to control noise amplification and potential multivalued solutions.

