# OpenReview forum: "Topology-aware Neural Flux Prediction Guided by Physics"
_ICML.cc/2025/Conference — ICML 2025 poster_

### Official Review · Reviewer_4Q4k · 2025-03-10

**Overall Recommendation:** 2

**Summary:**

This paper proposes a topology-aware prediction framework that adopts explicit difference matrices that model directional gradients and incorporates implicit physical constraints to difference matrices which enhances the consistency with physical laws. Experiments on two real-world data demonstrate the effectiveness of this framework.

**Claims And Evidence:**

I think most of the claims are clear and convincing.

**Essential References Not Discussed:**

Most related works are included.

**Experimental Designs Or Analyses:**

Important experiment details such as the number of training steps and the precise learning rate schedule are missing. Furthermore, the implementation details, such as the number of message passing layers for PhyNFP and baselines, are missing. These raise doubts about the credibility.

**Methods And Evaluation Criteria:**

The authors’ comparison with extensive baseline models is a strong point. However, using less common datasets like River and Traffic may limit generalizability and reproducibility. I suggest that authors compare PhyNFP with baselines on mainstream datasets, e.g., CylinderFlow, Airfoil from [1], or Eagle from [2].

[1] Learning Mesh-Based Simulation with Graph Networks.
[2] Eagle: Large-Scale Learning of Turbulent Fluid Dynamics with Mesh Transformers.

**Other Comments Or Suggestions:**

N/A

**Other Strengths And Weaknesses:**

**Strengths**
- PhyNFP combines discretized difference matrices with physical constraints.

**Weakness**
- The need to design specific message-passing mechanisms tailored to different physical dynamics, based on their unique constraints, may reduce the model's generality. This limitation could hinder PhyNFP's applicability to broader or more diverse physical systems without significant customization.

**Questions For Authors:**

1.  How does PhyNFP, trained on a known physical system, perform on similar physical systems with different physical constraints? What is the transferability or generalization of this model in this situation?
2. Can you give detailed dataset descriptions such as the number of training sequences and testing sequences?
3. Can you provide detailed implementation of PhyNFP and baseline models?

**Relation To Broader Scientific Literature:**

The paper situates itself within the broader literature on learning physics dynamics with graph neural networks (GNNs). It mainly contributes to this area by combining discretized difference matrices with implicit physical laws to address limitations in enforcing global consistency in flow dynamics.

However, a more detailed discussion of PhyNFP’s scalability and efficiency compared to existing methods would strengthen its contribution and highlight its practical advantages.

**Theoretical Claims:**

Theoretical claims are sound.

---

> ### Author Rebuttal · Authors · 2025-03-31
>
> Thank you for your constructive comments and questions. We address them in a Q\&A format as follows.
>
> Q1: How does PhyNFP perform on similar physical systems with different constraints?
>
> A1: The PDEs adopted by PhyNFP (Saint-Venant (S-V) for hydrodynamics and Aw-Rascle (A-R) for traffic flow) are basic and general formulations applicable to diverse scenarios within river and traffic networks. For instance, the gravitational term in S-V equations, crucial for elevation variations, can be naturally attenuated for flat rivers by adjusting learned weights during training. This enables PhyNFP to adapt across varying physical constraints without structural changes. Moreover, our ablation study shows that PhyNFP is robust even without explicit PDE constraints, which confirms its generalizability by using the difference operators alone.
>
> Q2: Can you give detailed dataset descriptions such as the number of training sequences and testing sequences?
>
> A2: We have provided dataset descriptions in Section 4 (Datasets, page 5). In particular, for the river dataset derived from LamaH-CE, we used data from the period 2000–2017, where the data from the years of 2016 and 2017 were used as the test set, and the remaining years were used for training. We followed [1] to make this train/test split. In our revised manuscript and forthcoming code repository, we will include all dataset statistics and detailed train/test splits for all datasets used.
>
>
> Q3: Can you provide detailed implementation of PhyNFP and baseline models?
>
> A3: The implementation specifics of PhyNFP are presented in Section 3.3, including the construction of difference matrices $D_1$ and $D_2$ and the PDE integration mechanism. In our revised manuscript and subsequent code release, we will further provide complete architectural descriptions, hyperparameter settings, and detailed training settings. The baseline models used (GraphSAGE, GCN, GAT, GWN, MP-PDE Solver, MPNN, and GNO) are introduced in Section 4 (Competitors, page 6).
>
>
> Q4: Comparison with Mesh-Based Benchmarks
>
> A4: We clarify key differences between mesh-based methods and our graph-based approach:
>
> 1)  Applicability to sparse, irregular networks. Mesh-based PDE methods require structured grids with regular spacing and clear geometric coordinates to ensure numerical accuracy. This does not match our datasets, which are sparse and irregular (such as river and traffic networks). In contrast, PhyNFP uses topology-based difference matrices that work well on irregular graphs and naturally capture directional flow.
>
> 2) Modeling topological structure. Mesh-based methods focus mostly on spatial resolution, but they do not model topological effects like upstream-downstream structure. Our method is designed to capture these topological patterns, which are important for real-world networks. This kind of topological sensitivity is hard to achieve using standard mesh-based solvers.
>
>
> Q5: Efficiency / Scalability
>
> A5: Although computational efficiency is not our core research contribution, we add an experiment below to show that the runtime of our method exhibits sub-linear scaling with larger graph size. As shown in the table above, when the number of nodes increases from 31 to 358 (more than 10×), the runtime per epoch only doubles, demonstrating that computational complexity grows significantly slower compared to the upscaling of graph size.
>
> | Number of Nodes | Average Runtime per Epoch(s) |
> |-------------------------|-------------------------------------------|
> | 9                           | 57.4                                           |
> | 27                         | 61.5                                           |
> | 31                         | 62.2                                           |
> | 358                       | 129.2                                         |
>
> Table: Average Runtime per Epoch for Different Graph Structures
>
> Reference:
>
> [1] Nikolas Kirschstein and Yixuan Sun, The Merit of River Network Topology for Neural Flood Forecasting, ICML 2024.

---

### Official Review · Reviewer_Euys · 2025-03-12

**Overall Recommendation:** 4

**Summary:**

The paper proposes a PhyNFP framework that aims to improve GNNs for modeling flow dynamics
in directed graphs. The main hypothesis of the paper is that the directional insensitivity
of traditional GNNs and their inability to capture high-frequency components arise, because
GNNs inherently smooth out directional variations. This makes GNNs struggle in distinguishing
forward and reverse flows. PhyNFP integrates both explicit difference matrices and global
physical constraints to overcome these challenges, where
the former encodes local directional dependencies and the latter enforces consistency
with natural laws.  The framework is validated on two  real-world directed graph datasets,
including a water flux network and an urban traffic flow network, and the emprical results
demonstrate its superior performance over standard GNNs and PDE-based solvers.

**Claims And Evidence:**

The paper makes three key theoretical claims. First, it asserts that the standard message-passing mechanism in GNNs acts as a low-pass filter, suppressing high-frequency components that are crucial for capturing sharp transitions and localized changes in directed flow dynamics. The paper investigates and supports this claim by by formulating an inverse problem, where the task is to predict upstream conditions based on downstream observations. The rationale lies in the fact that this inverse setup presents an ill-posed learning setting, amplifying high-frequency components, which traditional GNNs fail to retain.


Second, the proposed PhyNFP introduces discretized difference matrix (DDM), which approximates spatial gradients and preserve high-frequency information. These matrices modify the adjacency structure of the graph, ensuring that message passing retains fine-grained directional details. The paper validates the effectiveness of discretized difference matrices through theoretical analysis.Using discrete-time Fourier transform, the authors show that the frequency response of the difference matrix operator is $I + \alpha D$, demonstrates that high-frequency components are selectively preserved, unlike conventional adjacency-based smoothing which diminishes them. In emprical study, the authors define Direction Sensitivity (DS) and RDS as the difference metrics in prediction error between the original graph (Forward Flow) and a graph with reversed edge directions (Reverse Flow). The higher DS scores of PhyNFP compared to all baseline models indicate that PhyNFP distinguishes between forward and reverse flows more effectively than standard GNNs, validating that DDMs capture directional dependencies.


Third,  PhyNFP integrates physical law constraints directly into the GNN training process, such as conservation of momentum (Saint-Venant equations for river networks) and mass conservation (Aw-Rascle equations for traffic networks). This ensures that predictions remain physically consistent, reducing the reliance on purely data-driven patterns and reinforcing the structural priors inherent in real-world flow systems. This claim is validated through various aspects. 1) In theory, the paper formulates domain-specific physical constraints using governing equations (SV or AR), both are discretized and incorporated as regularization terms into the GNN loss function. 2) In experiment, PhyNFP is compared against purely data-driven GNNs (GCN, GraphSAGE, GAT, GWN, MPNN) and graph-based PDE solvers (MP-PDE Solver, GNO), and the results show that PhyNFP consistently outperforms both categories, confirming that physics constraints enhance model accuracy beyond what data-driven learning can achieve alone. 2) In ablation study, the paper evaluates how error grows as the prediction horizon (lead time) increases. While all models exhibit increasing error over longer horizons, PhyNFP grows its error at a significantly lower rate than baselines, indicating that physics constraints improve long-term stability.

**Essential References Not Discussed:**

I find the following papers related and should not be missing:

[1] Li, Zongyi, Nikola Kovachki, Kamyar Azizzadenesheli, Burigede Liu, Kaushik Bhattacharya, Andrew Stuart, and Anima Anandkumar. "Neural operator: Graph kernel network for partial differential equations." arXiv preprint arXiv:2003.03485 (2020).

[2] Karniadakis, George Em, Ioannis G. Kevrekidis, Lu Lu, Paris Perdikaris, Sifan Wang, and Liu Yang. "Physics-informed machine learning." Nature Reviews Physics 3, no. 6 (2021): 422-440.

[3] Dong, Yushun, Kaize Ding, Brian Jalaian, Shuiwang Ji, and Jundong Li. "Adagnn: Graph neural networks with adaptive frequency response filter." In Proceedings of the 30th ACM international conference on information & knowledge management, pp. 392-401. 2021.

**Experimental Designs Or Analyses:**

The experiments are set up in stardard and structured ways. Two real directed graphs are employed for benchmark, which makes the proposed framework executable compared to the PINN studies that use simulation only. The proposed PhyNFP is compared against traditional GNN models and well as graph-based PDE solvers. Supervised node regression to predict flux volume at a future time step is used as the benchmark task, and model accuracy is assessed using MSE.

A crucial aspect of the experimental setup is the direction sensitivity analysis. The authors evaluate models in both the original graph setting (Forward Flow) and an inverse setting (Reverse Flow), whereby using DS and RDS to quantify how much the prediction error changes. The positive results of PhyNFP demonstrates its effectiveness and verifies the tightness of its technical findings.

**Methods And Evaluation Criteria:**

The proposed method is technically sound, with clear notations following many domain conventions, with well-defined optimization objectives. The use of discretized difference matrices to construct new adjacency operators aligns with numerical methods for hyperbolic PDEs, ensuring that information flow follows physically meaningful gradients, is novel.

The evaluation metrics are appropriate, with MSE as the primary metric for predictive accuracy and DS and RDS for assessing directional awareness. The inclusion of baseline comparisons, ablation studies, and robustness evaluations strengthens the validity of the results.

**Other Comments Or Suggestions:**

Provide a runtime and complexity analysis to evaluate the scalability of PhyNFP.
Conduct feature visualization to analyze how the difference matrices and PDE constraints influence learned representations.
Explore applications beyond flux prediction, such as graph-based anomaly detection or turbulent flow modeling.

**Other Strengths And Weaknesses:**

A key strength of the paper is its well-motivated problem formulation and strong empirical validation. The integration of numerical methods (difference matrices) and physics principles is novel and well-executed. Additionally, the direction sensitivity evaluation is an important contribution to the study of GNNs for flow-based systems.

However, the paper has a few weaknesses. First, the computational efficiency is not analyzed in detail, raising concerns about its applicability to large-scale graphs. Second, the interpretability of the learned embeddings is not discussed—how do the physics constraints influence the feature representations in the GNN layers? Finally, the framework is evaluated only on node regression tasks; it would be valuable to explore whether it can generalize to link prediction or spatio-temporal forecasting.

**Questions For Authors:**

1. How does PhyNFP scale to large networks and what are its computational bottlenecks?
2. Can this framework be applied to link prediction tasks in dynamic graphs?
3. Is PhyNFP robust to incomplete or noisy data? How does it handle missing node attributes?
4. What does the "undirected" in Figure 1 mean and how it was implemented?

**Relation To Broader Scientific Literature:**

The paper is likely to enjoy a broader audience group by situating its contributions within GNNs for physical systems. It discusses graph-based PDE solvers, physics-informed neural networks, and spatio-temporal GNNs for flood and traffic prediction. However, it could provide a stronger comparison with recent works on hybrid GNN-PDE models, such as Neural Operators and Physics-Guided Neural Networks.

**Theoretical Claims:**

The paper makes three key theoretical claims. First, it asserts that the standard message-passing mechanism in GNNs acts as a low-pass filter, suppressing high-frequency components that are crucial for capturing sharp transitions and localized changes in directed flow dynamics. The paper investigates and supports this claim by by formulating an inverse problem, where the task is to predict upstream conditions based on downstream observations. The rationale lies in the fact that this inverse setup presents an ill-posed learning setting, amplifying high-frequency components, which traditional GNNs fail to retain.


Second, the proposed PhyNFP introduces discretized difference matrix (DDM), which approximates spatial gradients and preserve high-frequency information. These matrices modify the adjacency structure of the graph, ensuring that message passing retains fine-grained directional details. The paper validates the effectiveness of discretized difference matrices through theoretical analysis.Using discrete-time Fourier transform, the authors show that the frequency response of the difference matrix operator is $I + \alpha D$, demonstrates that high-frequency components are selectively preserved, unlike conventional adjacency-based smoothing which diminishes them. In emprical study, the authors define Direction Sensitivity (DS) and RDS as the difference metrics in prediction error between the original graph (Forward Flow) and a graph with reversed edge directions (Reverse Flow). The higher DS scores of PhyNFP compared to all baseline models indicate that PhyNFP distinguishes between forward and reverse flows more effectively than standard GNNs, validating that DDMs capture directional dependencies.


Third,  PhyNFP integrates physical law constraints directly into the GNN training process, such as conservation of momentum (Saint-Venant equations for river networks) and mass conservation (Aw-Rascle equations for traffic networks). This ensures that predictions remain physically consistent, reducing the reliance on purely data-driven patterns and reinforcing the structural priors inherent in real-world flow systems. This claim is validated through various aspects. 1) In theory, the paper formulates domain-specific physical constraints using governing equations (SV or AR), both are discretized and incorporated as regularization terms into the GNN loss function. 2) In experiment, PhyNFP is compared against purely data-driven GNNs (GCN, GraphSAGE, GAT, GWN, MPNN) and graph-based PDE solvers (MP-PDE Solver, GNO), and the results show that PhyNFP consistently outperforms both categories, confirming that physics constraints enhance model accuracy beyond what data-driven learning can achieve alone. 2) In ablation study, the paper evaluates how error grows as the prediction horizon (lead time) increases. While all models exhibit increasing error over longer horizons, PhyNFP grows its error at a significantly lower rate than baselines, indicating that physics constraints improve long-term stability.

---

> ### Author Rebuttal · Authors · 2025-03-31
>
> Thank you for your constructive comments and questions. We address them in a Q&A format as follows.
>
> Q1: How does PhyNFP scale to large networks and what are its computational bottlenecks?
>
> | Number of Nodes | Average Runtime per Epoch(s) |
> |-------------------------|-------------------------------------------|
> | 9                           | 57.4                                           |
> | 27                         | 61.5                                           |
> | 31                         | 62.2                                           |
> | 358                       | 129.2                                         |
>
> Table: Average Runtime per Epoch for Different Graph Structures
>
> A1: We add an experiment to show that the runtime of our method exhibits excellent sub-linear scaling with larger graph size. As shown in the table above, when the number of nodes increases from 31 to 358 (more than 10×), the runtime per epoch only doubles, demonstrating that computational complexity grows significantly slower than the graph size.
>
>
> Q2: Can this framework be applied to link prediction tasks in dynamic graphs?
>
> A2: Our framework is designed for flux prediction in physical systems, focusing on modeling directional flows governed by physical laws. Link prediction in dynamic graphs is a structurally different task, aiming to infer future or missing edges. Such problems are rarely encountered in our physical settings. We would like to defer this interesting setup by adapting our method for dynamic link prediction as a future work.
>
>
> Q3: Is PhyNFP robust to incomplete or noisy data? How does it handle missing node attributes?
>
> A3: Yes, our model is robust to both incomplete and noisy data. Our datasets include real-world measurement data, which naturally contain both noisy and missing entries. Our method can mitigate the effect of noise through its physics-guided inductive bias, which acts as implicit regularization to enhance robustness.
>
> For missing node attributes, we adopt the preprocessing step used in~[1], where nodes with missing features are excluded from the network.
>
> Q4: What does the "undirected" in Figure 1 mean and how it was implemented?
>
> A4: In the “undirected” setting we remove edge directions by symmetrizing the adjacency matrix (i.e., using $A_{\text{undirected}} = A + A^\top$), so each node aggregates messages from its upstream and downstream neighbors. In this setting, the model does not differentiate between upstream and downstream connections. In contrast, the forward setting aggregates only from upstream neighbors, and the reverse setting flips all edge directions, so aggregation only considers those originally downstream nodes.
>
> [1] Nikolas Kirschstein and Yixuan Sun, The Merit of River Network Topology for Neural Flood Forecasting, ICML 2024.

---

### Official Review · Reviewer_b6U4 · 2025-03-14

**Overall Recommendation:** 2

**Summary:**

This paper addresses the challenge of preserving high-frequency components in Graph Neural Networks (GNNs) when applied to directed graphs, which is crucial for accurately modeling flow dynamics. Traditional GNNs often fail to distinguish between forward and reverse graph topologies, leading to information loss. To overcome this, the authors propose a framework that integrates explicit difference matrices for modeling directional gradients and implicit physical constraints to ensure message passing aligns with natural laws. Experiments on real-world datasets, including water flux and urban traffic networks, demonstrate the effectiveness of the proposed approach.

**Claims And Evidence:**

Not really. The proposed method's improvement is not as significant as the authors claim. For instance, its performance on the reverse model is worse than some baselines. Additionally, the reported 4.9% performance gain on the river dataset is misleading, as it is calculated by averaging the forward MSE error across all methods rather than making a fair one-to-one comparison. When compared to the second-best model, the actual improvement is only around 0.5%, which is minimal. Furthermore, since the experiments were conducted using only a single seed, this minor performance gain may not hold when averaged over multiple runs.

**Essential References Not Discussed:**

The authors provide a relatively comprehensive discussion of related work. However, methods like PINN-GNN approaches could be included. This is because the selected graph learning models for physical systems do not explicitly incorporate physical laws. Research on enforcing physical constraints through loss functions (for example) is relevant to the problem and should be considered for comparison with the proposed method.

**Experimental Designs Or Analyses:**

As mentioned earlier, the way that the authors compared the performance of the proposed method is unfair. The metrics is also problematic and seems not be able to well justify or measure the direction sensitivity. Moreover the proposed method has worse reverse MSE under traffic network dataset, which is not discussed in the paper. More detailed questions will be mentioned below.

**Methods And Evaluation Criteria:**

The evaluation criterial seems problematic. The authors defined direction sensitivity of a certain model $M$ as $DS(M)$ =
$l_M$(Reverse)−$l_M$(Forward), where $ℓ_M$ indicates the MSE loss of M. According to the paper, a higher $DS$ value indicates a better model, but this is misleading. A model with poor performance on $l_M$(Reverse) (i.e. high MSE error) would naturally have a larger DS, which does not necessarily reflect improved direction sensitivity.

**Other Comments Or Suggestions:**

Please refer to the question section.

**Other Strengths And Weaknesses:**

The idea of modifying the adjacency matrix and update function to incorporate physical constraints is intriguing. However, the results show that the proposed method does not consistently outperform other approaches on certain datasets. The authors do not provide clear explanations for this, and the proposed evaluation metrics appear problematic in accurately assessing the method’s performance. Moreover, it is questionable on how generalizable is this approach to other system that cannot be discretized in similar format.

**Questions For Authors:**

1. It is unclear why, in the reverse problem, small numerical errors in the inference process propagate and amplify, leading to instability and sensitivity in the reconstructed upstream conditions. Could the authors clarify why this issue occurs specifically in the reverse problem and whether it could also be a concern for the forward problem?
2. The generalizability of this approach to systems that cannot be discretized in a similar format is questionable. Could the authors discuss its applicability to broader cases?
3. The derivation of Eq. (12) and Eq. (13) from Eq. (8) and Eq. (11) is unclear.For instance, what happens to the term $\rho^n$ in Eq. (13)? Additional details on the derivation would be helpful.
4. The intuition behind Eq. (4) is not well explained. It seems like D1 and D2 are introduced primarily to facilitate the derivation of Eq. (12) and Eq. (13). Could the authors clarify their role?
5. The text immediately following Eq. (11) does not seem to align with the equation.
6. The authors mentioned $\Delta t$ and $\hat g$ are learnable scalars. How is the accuracy or reliability of these learned terms verified? 7. The authors mentioned *"By making $\Delta t$ learnable, GNNs can adjust their sensitivity to real-time traffic conditions, providing a physics-aware approach to traffic prediction"*. Could the authors provide the actual learned values of  $\Delta t$ ? Do these values make sense, and do they adapt to different conditions as expected? Additionally, how would the performance compare if a fixed $\Delta t$ were used instead?
8. How does the model handle input features with 24 time steps? Is it through concatenation or another method?
9. Regarding the concerns mentioned about the $DS$ metric, could the authors clarify if there is any misunderstanding?
10. In Figure 2(c), the method is compared with ResGCN, which was not included in previous baseline comparisons. Additionally, since GCN performs the worst in the forward model, why not compare it with the second-best model (e.g., GWN)? Furthermore, how does the performance differ for evaluation in RQ4 under the traffic network setting? Also, what happens when nodes are perturbed in the reverse model?
11. Can the authors explain why the proposed method underperforms in the reverse model on the traffic network dataset?
12. The paper lacks sufficient details on the model architecture and training settings, making reproduction difficult.

**Relation To Broader Scientific Literature:**

The paper enhances GNNs for directed graphs by preserving high-frequency components crucial for flow dynamics modeling. It builds on spectral graph theory and physics-informed learning, introducing directional gradients and physical constraints.

**Theoretical Claims:**

It is unclear how Eq12 and Eq13 are derived.

---

> ### Author Rebuttal · Authors · 2025-03-31
>
> Thank you, and we address your questions in a Q\&A fashion as follows
>
> Q1: Why the reverse problem will amplify numerical errors? Is it a concern for the forward problem?
>
> A1: Solving inverse problems is ill-posed, as many different upstream conditions can lead to the same downstream fluxes. When inferring upstream states from downstream inputs, the mapping becomes one-to-many and numerically unstable, small noise at downstream nodes can cause large errors upstream. In our inverse experiment (Q9), we observe that GCN suppresses such noise entirely, showing no upstream variation. This supports our hypothesis that GCN cannot distinguish forward from reverse settings and lacks sensitivity to edge direction. Further details are provided in our response to Q9.
>
> This is not a concern in the forward setting because the PDE describing physical structure can enforce a stable and one-to-one mapping from upstream to downstream nodes.
>
> Q2: Generalize to broader systems cannot be discretized?
>
> A2: Our discretization does not rely on any specific PDE; rather, it is based on the topological structure describing the system. In fact, we can construct multiple difference operators [1] from the graph through its spatial or functional adjacency, and our difference matrix derived from local spatial variations and directional flow is one possible instantiation.
>
> Q3: How were Eqs.(12) and (13) derived from Eqs.(8) and (11)?
>
> A3: We will supplement a step-by-step derivation, as presented in (https://anonymous.4open.science/r/PhyNFP-D88F/Q3.pdf). The main idea is to use independent MLPs that take raw node inputs to learn representations of physical quantities. Difference matrices $D_1$ and $D_2$ are applied to guide the representation learning, so to align with the original PDE. The intuitions behind $D_1$ and $D_2$ are in our response to Q4.
>
> Q4: Are D1 and D2 just introduced to enable Eqs.(12) and (13) derivation?
>
> A4: $D_1$ and $D_2$ are not just for derivation, they capture key physical quantities. $D_1$ models horizontal gradients on the graph, reflecting convection and spatial variation. $D_2$ captures vertical gradients from elevation, tied to gravity-driven flow. These matrices allow our framework to generalize to other PDE systems involving similar terms.
>
> Q5: The text following Eq. (11) does not with the equation.
>
> A5: We will fix this typo, as also suggested by Reviewer RhQb.
>
> Q6: Why is $\Delta t$ (and $\hat{g}$) set as learnable scalar but not fixed?
>
> A6:  The amount of $\Delta t$ determines either a message-passing layer will update the node embeddings by reusing the output from previous layer (small $\Delta t$) or it will incorporate the information for updating the physical status of PDE (large $\Delta t$). For example, a large $\Delta t$ in Eq.(12) means that the embedding update will mostly rely on using the convection and gravity terms derived from the Eq.(6). By learning it from data, we make $\Delta t$ adaptable to real conditions.
>
> We validate through two experiments: 1) Larger $\Delta t$ improves $DS$, as shown in Table 1 in the anonymous link ( https://anonymous.4open.science/r/PhyNFP-D88F/README.md), although inverse problems typically prefer smaller steps [2]. With fixed layers, small $\Delta t$ limits each update. A learnable $\Delta t$ adapts to network depth without this concern. 2) Figure 2 in the link plots the variation of $\Delta t$ w.r.t. the number of epochs, where it converges to small values in the reverse setting.
>
> Q7: Handle input features with 24 time steps?
>
> A7: Concatenation. Please refer to our response to Q1 of Reviewer RhQb.
>
> Q8: Misunderstanding in DS?
>
> A8: Indeed a high reverse MSE can produce high DS performance. However, our model enjoys the highest DS = +.0105 by attaining the lowest reverse MSE = .0906. The positive and high DS value indicates a better distinguishability of edge directions.
>
> Q9: 1) Why ResGCN is not in Table 1? 2) How perturbations reflect in reverse setting?
>
> A9: Answer to 1) is in Q3 of Reviewer RhQb.
>
> To answer 2),  we add local perturbation in reverse setting (and the Traffic Network) as shown in Fig 1 (and Fig 3) in the anonymous link.
>
> Q10: Performance downgrade in traffic network?
>
> A10: Mainly due to cycles in the traffic network. Using Johnson's algorithm, we find 80 cycles in it, but the river network has zero. These cycles allow messages to propagate in both directions, making it more challenging for models to distinguish forward and reverse flows.
>
> Q11: Details on the model architecture and training settings?
>
> A11: We use 19 layers based on the graph diameter in our dataset. In experiment we observe that our implementation is robust across different layer numbers. We will publish our code with all hyperparameter settings for reproducibility.
>
> [1] Grady, et al. Discrete calculus: Applied analysis on graphs for computational science[M]. London: Springer, 2010.
>
> [2] Bertero, et al. The stability of inverse problems[M]. Springer Berlin Heidelberg, 1980.

---

### Official Review · Reviewer_RhQb · 2025-03-27

**Overall Recommendation:** 4

**Summary:**

The authors proposed PhyNFP, a topology-aware neural flux prediction framework that integrates GNNs with physical principles to improve flow dynamics modeling in directed graphs. Traditional GNNs struggle with directional sensitivity and high-frequency information loss due to their inherent low-pass filtering nature. PhyNFP addresses this limitation by incorporating difference matrices that encode local directional dependencies and global physical constraints that ensure predictions adhere to real-world physics. The authors evaluate PhyNFP on two real-world datasets, where it outperforms traditional GNNs and PDE-based solvers in terms of both accuracy and directional sensitivity.

**Claims And Evidence:**

The claims are legitimate and supported with sufficient empricial evidence.

**Essential References Not Discussed:**

I do not find significant missing.

**Ethical Review Concerns:**

None noted.

**Experimental Designs Or Analyses:**

The paper employs a structured and rigorous experimental setup, using two real-world directed graph datasets. PhyNFP is compared against standard GNNs and PDE-based solvers. The competitors chosen can represent the state of the art. Standard metrics like MSE are used as it being widely employed in other node-level regression tasks. Ablation studies confirm that both difference matrices and physics constraints are essential, while removing them leads to higher errors and reduced stability. Long-horizon prediction tests show PhyNFP accumulates less error over time.

**Methods And Evaluation Criteria:**

The proposed method is rigorous yet straightforward, which is good. The presentation makes it easy to follow. The evaluation strategy is standard and thorough.

**Other Comments Or Suggestions:**

Please find my comments above.

**Other Strengths And Weaknesses:**

+ The study is serious, and its proposed framework is straightforward but can be generalized into various related domains like power grids, epidemiology, and financial transaction networks where edge directions are critical.

+ The design of using difference matrix and combining it with PDEs to design new message-passing layers is novel and easy to implement. I am in favor of this explicit and deterministic designs over inplicite regularizations.

+ The analysis in appendix followed the standard steps in graph spectral analysis and and results support its legtimacy.

+ The evaluation in real datasets rather than those from simulators are extensive and positive. I appreciate more such PINN research to be evaluated on real datasets.

**Questions For Authors:**

- How was exactly the flux prediction or regression task modeled? Is it autoregressive for the past 24 time steps? If so, what is the base learner?

- I believe that the notations in Eq.11 and their elaborations do not match. Please make it consistent and explain which is which.

- What is the main finding of Fig.2? Why ResGCN is good/bad given its reaction to the "change" (what is thischange?) in v_1? Why ResGCN is not included as one of the GNN competitors?

- How robust is PhyNFP to data incompleteness? Given the two datasets collected through observatories, there may have missing entries -- how were those handled?

- What do you mean by forward vs reverse vs undirected?

**Relation To Broader Scientific Literature:**

The paper situates its contributions within physics-informed machine learning and graph learning, drawing connections to spatio-temporal GNNs and numerical solvers for flow dynamics. It highlights the limitations of standard GNNs in handling directional dependencies and emphasizes the need for physics-guided regularization. The paper is likely to host audiences from a diversity of backgrounds.

**Theoretical Claims:**

I note two main claims made in this paper:

1) The authors argue that GNNs act as low-pass filters, suppressing high-frequency components critical for flow directionality. This makes them insensitive to forward vs. reverse flows.

The claim is supported by Fourier analysis (in appendix), showing that message-passing smooths signals, and an inverse problem formulation, which demonstrates that predicting upstream flux from downstream conditions is ill-posed. The DS and RDS metrics further confirm that standard GNNs fail to capture directionality, while PhyNFP excels.

2) The authors claim that incorporating physical constraints improves model accuracy and stability by embedding Saint-Venant (hydrology) and Aw-Rascle (traffic) equations into the loss function. The theoretical formulation ensures physics-consistent learning, while ablation studies confirm that removing constraints significantly degrades accuracy. Also, long-horizon experiments show that PhyNFP accumulates less error over time, proving its stability advantage.

I identify these claims to be legitimate and likely to contribute to the advancement of physics-informed GNNs for flow-based systems.

---

> ### Author Rebuttal · Authors · 2025-03-31
>
> Thank you for your constructive comments and questions. We address them in a Q&A format as follows.
>
> Q1: How was the flux prediction task modeled? Is it autoregressive for the past 24 time steps?
>
> A1: The flux prediction task is formulated as a supervised node regression problem rather than an autoregressive one. For each node, a fixed-length temporal window of 24 past time steps \(W=24\) is concatenated into one feature vector, following the prior research~[1]. This vector is used to predict the future flux \(y\) at time \(t+6\). The base learner is graph neural network that integrates discretized difference matrices and physical priors in its message passing process.
>
>
> Q2: Notation inconsistency in Eq.(11) and their elaborations.
>
> A2: This is a typo, and we will make revision in our camera-ready as follows:
> $$
> \rho_i^{t+1} = \rho_i^t - \alpha \left( \rho_i^t (\hat{D} u^t)_i + u_i^t (\hat{D} \rho^t)_i \right),
> $$
>
> where $(\hat{D}u^t)_i$ represents velocity differences, and $(\hat{D}\rho^t)_i$ encodes density-driven effects.
> These terms approximate the spatial derivatives of velocity and density, respectively, using a difference operator $\hat{D}$. The scalar factor $\alpha$ is defined as $\Delta t / \Delta x$.
>
>
> Q3: What is the main finding of Fig.2? Why is ResGCN good/bad given its reaction to the "change" (what is this change?) in v-1? Why is ResGCN not included as one of the GNN competitors?
>
> A3: The "change" refers to a manually injected spike in the input flux at node $v_1$, simulating a sudden perturbation in the river network. Fig.2 examines how this perturbation propagates downstream through the model predictions. The full analysis of the main finding in Fig.2 is provided in RQ4 of the main text. We will further clarify in the final version.
>
> The result of ResGCN in Figure 2 is bad compared to our method because it fails to capture the correct flow dynamics. Specifically, when the perturbation is applied to a node, it should cause an increase in its downstream flux, while GCN shows almost no change in downstream nodes. Although ResGCN propagates this perturbation signal to some extent, it produces incorrect trends; for example, predicting a decrease at node \(v_2\) where an increase is expected.
>
> We select GCN due to its wider application over its residual variant. In fact,  ResGCN exhibits similar performance to standard GCN with shallow layers. Even after careful tuning (e.g., adding more layers), ResGCN only performs on a par with GWN by having MSE (F) = .1114, MSE (R) = .1139, and RDS = -76.2\%. We shall include those results in our camera ready.
>
>
> Q4: How robust is PhyNFP to data incompleteness and how are the missing entries handled?
>
> A4: Our datasets include real-world measurement data, which naturally contain both noisy and missing entries. Our method can mitigate the effect of noise through its physics-guided inductive bias, which acts as implicit regularization to enhance robustness.
>
> For missing node attributes, we adopt the preprocessing step used in~[1], where nodes with missing features are excluded from the network.
>
> Q5: What are the settings of forward vs reverse vs undirected?
>
> A5: In our work, these terms describe different ways of using edge directions during graph construction and message passing. In the forward setting, we use the original directed graph where edges follow the real physical flow (i.e., from upstream to downstream). This reflects the correct direction of information propagation.
>
> In the reverse setting, all edge directions are flipped (i.e., the adjacency matrix is transposed), so information flows in the opposite direction (i.e., from downstream to  upstream), which violates the physical rules.
>
> In the undirected setting, edge directions are ignored. Each node exchanges messages with both upstream and downstream neighbors, which is common in standard GNNs.
>
> Reference:
> [1] Nikolas Kirschstein and Yixuan Sun, The Merit of River Network Topology for Neural Flood Forecasting, ICML 2024.

---

### Decision · Program_Chairs · 2025-05-01

**Decision:**

Accept (poster)

**Comment:**

This paper proposes a framework called PhyNFP to enhance graph neural networks (GNNs) for flux prediction on directed graphs. Its advantages include: (1) accurately modeling directionality by distinguishing forward vs. reverse flows; (2) predictions remain physically plausible by embedding domain-specific physical laws; (3) preserving high-frequency signals, allowing better modeling of sharp transitions and local phenomena; (4) maintaining better performance under data noise and long-term prediction. Concerns mostly lie in marginal gains in some corner cases, inconsistencies in reverse setting, and lack of interpretability of the learned embeddings.